# Intestinal Microbiota in Children with Anemia in Southern Peru through Next-Generation Sequencing Technology

**DOI:** 10.3390/children9111615

**Published:** 2022-10-25

**Authors:** Karla Díaz-Rodríguez, Jani Pacheco-Aranibar, Cecilia Manrique-Sam, Yuma Ita-Balta, Agueda Muñoz del Carpio-Toia, Patricia López-Casaperalta, Teresa Chocano-Rosas, Fernando Fernandez-F, Jose Villanueva-Salas, Julio Cesar Bernabe-Ortiz

**Affiliations:** 1Post-Graduate School, Universidad Católica de Santa María, Urb. San José s/n, Umacollo, Arequipa 04013, Peru; 2Deparment of Biology, Universidad Nacional de San Agustín, Santa Catalina Nro. 117, Arequipa 04001, Peru; 3Department of Molecular Biology, Instituto de Biotecnología del ADN Uchumayo, Arequipa 04401, Peru; 4Vicerrectorado de Investigación, Universidad Católica de Santa María, Urb. San José s/n, Umacollo, Arequipa 04013, Peru

**Keywords:** 16S rRNA, intestinal microbiota, anemia

## Abstract

Knowledge of the sequencing of the 16S rRNA gene constitutes a true revolution in understanding the composition of the intestinal microbiota and its implication in health states. This study details microbial composition through next-generation sequencing (NGS) technology in children with anemia. Anemia is the most frequent hematological disorder that affects human beings. In Peru, it is one of the conditions that presents the most significant concern due to the adverse effects that cause it, such as delayed growth and psychomotor development, in addition to a deficiency in cognitive development.

## 1. Introduction

Recent studies indicate that iron deficiency (ID) [1] and iron deficiency anemia (IDA) cause unfavorable changes in the intestinal microbiota [2]. There is an excellent relationship between microorganisms of the small intestine and living beings, beginning at birth and continuing throughout life [3,4].

To prevent anemia, the diet can be supplemented with iron; however, certain bacteria are efficient iron scavengers [5]. Secondly, the amount of iron available in the intestine influences the microbiota of infants, which can be highly mutable [6].

The gastrointestinal tract (GIT) contains approximately 70% of the total microorganisms in the body. Due to its temperature and the amount of nutrients is contains, GIT is one of the preferred sites for the proliferation of microorganisms [1]. Low iron conditions, such as excess iron, show adverse effects, suggesting that there may be an optimal range of iron in the gut that directly affects the amount and variety of gut microbiota [7]. All microorganisms require iron to survive, except for *Lactobacilli* and *Borrelia burgdorferi* [8].

In a study carried out in 2019 by the INEI and the Ministry of Economy and Finance of Peru [9], it was observed that the prevalence of anemia in boys and girls under three years of age was different according to the regions of the country. Thus, the prevalence was high in the Sierra (48.8%) and Selva (44.6%) regions, decreasing to 33.9% in the Coast region.

The diversity of sequences, which provides the differentiation between bacterial species, varies according to the 16S rRNA gene. Nine regions of the 16S rRNA gene of high sequence variability have been identified (named hypervariable V1–V9 regions [10,11]). The present study seeks to identify intestinal microbiota in children with anemia from south Peru using next-generation sequencing (NGS) technology.

## 2. Materials and Methods

The cross-sectional study included 18 children from southern Peru under ten years of age. The patients were divided into three groups, two according to the blood hemoglobin level, and a control group. The first group consisted of ten children with anemia (AA) whose hemoglobin value was less than 11 g/dL. The second group consisted of four children who recovered from anemia (RA), and whose hemoglobin values were greater than 11 g/dL. Finally, the control group comprised four children who did not present anemia (C). For all cases, parental consent was obtained before the study, and the research was endorsed by the ethics committee of the Universidad Católica de Santa María (DICTAMEN 196-2020, on 21 December 2020) in Arequipa, Peru. The collection of samples consisted of obtaining feces kept in airtight jars (Medical Wire & Equipment Co., Corsham, UK), keeping them cold for conservation.

### 2.1. Microbial DNA Extraction

The DNA extraction process was divided into two stages, a purification process and an extraction process. In the purification stage, 1000 mg of the sample was diluted in 3 mL of 0.9% NaCl in a 15 mL falcon tube and vortexed (Eurolab, Madrid, Spain) for 30 s. Next, the solution was transferred to a roller shaker (JP Selecta, Barcelona, Spain) for ten minutes, and, once finished, it was centrifuged at 3000× *g* rpm for two minutes (Premiere-XC-2450, Philadelphia, PA, USA). Next, the supernatant was kept in microcentrifuge tubes (Eppendorf^TM^, Madrid, Spain) to be centrifuged at 10,000× *g* rpm for one minute in a microcentrifuge (Dlab Scientific, Beijing, China), keeping the pellet. To the latter, 1 mL of saline phosphate buffer (PBS) was added, centrifuging again at 3000× *g* rpm for two minutes. The supernatant was centrifuged at 10,000× *g* rpm for one minute, and the resulting pellet was preserved. Finally, in the extraction stage, the technique of phenol, chloroform, and isoamyl alcohol was used in a 24:25:1 ratio.

### 2.2. Sequencing of the V3-V4 Region of the 16S rRNA Gene

Sequencing libraries were generated using the NEBNext^®^ Ultra^TM^ DNA Library Pre Kit for Illumina, following the recommendations of the manufacturer, and index codes were added. Library quality was assessed using the Qubit^®^ 2.0 fluorometer (Thermo Scientific, Waltham, MA, USA) and the Agilent Bioanalyzer 2100 system. Finally, the library was sequenced on an Illumina platform, and 250 bp paired-end reads were generated.

### 2.3. Bioinformatic Analysis

The paired-end reads were merged using FLASH v.1.2.7, a fast computational tool able to extend short read lengths generated by NGS technologies. Quality filtering on the raw labels was performed under specific filtering conditions to obtain high-quality clean labels according to the QIIME quality control process. The UCHIME algorithm was used to detect chimeric sequences and remove them. Sequence analysis was performed with the Uparse software (Uparse is implemented as a command in USARCH, version is 11.0.667). For each representative sequence, the GreenGene database was used to study the phylogenetic relationship of different operational taxonomic units (OTUs) and the difference between the dominant species in different samples (groups). Sequence alignment was performed with the MUSCLE software (MUSCLE is available as a free web service on the EBI website and does not contain a version number). Alpha diversity and Beta diversity in weighted and unweighted UniFrac were calculated using QIIME software (QIIME 2 q2studio-2022.8.0).

### 2.4. Statistical Analyses

Statistical analyzes were performed with R software version 2.15.3, using the FactoMineR package (Vienna, Austria) and ggplot2 for principal component analysis (PCA) and cluster analysis. Principal coordinate analyses (PCoA) were performed with the WGCNA, stat, and ggplot2 packages included in the R software, using weighted and unweighted UniFrac distances. The linear discriminant analysis effect size was obtained with the LEfSe software (this is available as a module within MicrobiomeAnalyst which is a free web-based tool and does not contain a version number). The *p*-value was calculated using the permutation test method, while the q-value was calculated using the Benjamini and Hochberg false discovery rate method. Anosim, MRPP, and Adonis analyses were performed with the vegan R package. AMOVA was calculated using the open code mothur software (version 1.48.0.) with the amova function. Finally, the *t*-test was conducted with R software.

## 3. Results

### 3.1. Predominant Phyla and Genera Identified

Figure 1 shows the OTU annotation circular tree diagram of the group of anemic children. For this purpose, the GraPhIAn software tool was used. This software groups the effective tags of the samples according to the percentage of identity to obtain the OTUs. In this work, the percentage used for the tree diagram is 97% of identity. The phyla present in the group are Actinobacteriota, Bacteroidota, Firmicutes, and Proteobacteria.

### 3.2. Relative Abundance of the Main Phyla

The samples of the three groups studied (AA, RA, and C) were analyzed to obtain the distribution abundance of the intestinal microbiota. Figure 2 shows the relative abundance of the top ten phyla (*y*-axis) against the sampled groups (*x*-axis). The results show a higher relative abundance of the *Firmicutes*, *Actinobacteriota*, *Bacteroidota*, and *Proteobacteria phyla*. At the same time, representative populations are found for the bacteria *Verrucomicrobiota*, *Acidobacteriota*, *Cyanobacteria*, *Fusobacteriota*, *Synergistota*, and *Chloroflexi*. In addition, other phyla are found that do not show a representative percentage.

### 3.3. Shared and Unique OTUs

According to the analysis results of OTUs grouping and the investigation requirements, the table of the OTUs was normalized. Shared and unique information was analyzed for the different samples (groups) studied, and Venn and Flower diagrams were constructed. Appendix A shows the Venn diagram of the control group. Each circle represents the OTUs obtained in each sample. The values in the overlapping zones represent the common OTUs, while the remaining values are the unique OTUs. Each sample or individual of this group is labeled (A2, A3, A7, and A12). Thus, for A2, 416 unique OTUs are found; for A3, 87; for A7, 659; and for A12, 316 unique OTUs. For the four control samples, 160 common OTUs are found. Appendix A shows the Venn diagram for the samples of recovered children. For this group, the samples are labeled A8, A9, A14, and A15. The analysis yields the following data for unique OTUs: A8 with 110, A9 with 253, A14 with 115, and A15 with 113. The four samples share 296 OTUs. Appendix A shows the samples from children with iron deficiency anemia (IDA). Each petal in the flower diagram represents a sample and is differentiated with different colors. The central number (core 40) is for the number of OTUs present in all samples, while the number in the petals is for the unique OTUs found in each sample.

### 3.4. Significant Difference between AA and RA Groups

Figure 3 compares the groups of children with anemia (AA) and those who recovered (RA). The left panel represents the abundance of the *Clostridia* class and the orders *Peptostreptococcales-Tissierellales* and *Clostridiales*. The average value of the abundance of these taxonomic categories is represented by bars, whose color identifies each group. It can be seen that the *Clostridia* class and *Peptostreptococcales-Tissierellales* order are more abundant in the RA group, while the *Clostridiales* order is more numerous in the AA group. The confidence interval of variation between both groups is represented in the right panel. The lower and upper limits symbolize the 95% confidence interval, while the circle is the mean value. The circle color agrees with the group whose mean value is higher. On the *y*-axis, the *p*-value of the test of significance of group variation is shown.

### 3.5. Validity of the Sequencing Data: Rarefaction Curve

Rarefaction curves and rank abundance curves are widely used to indicate the biodiversity of samples. Appendix A shows the rarefaction curves obtained from all the samples analyzed. These curves are created by randomly selecting a certain amount of sequencing data from the samples, and then counting the number of species they represent (i.e., the number of OTUs). Rarefaction curves can directly reflect the rationality of the sequencing data volume and indirectly reflect the microbial community’s richness in the samples. Results show that the A14 and A15 curves (children recovered from anemia) are pronounced, suggesting many species remain to be discovered in these samples. On the other hand, the curves for samples A6 and A1 (children with anemia) become flatter, implying that a credible number of samples have been taken, meaning that only rare species remain to be sampled.

### 3.6. Alpha Diversity of the Gut Microbiota

Alpha diversity was quantified by the Shannon diversity index, which relates to both OTU richness and uniformity, and by the total number of species observed. Figure 4 shows alpha diversity measures for the AA group and children of the RA group compared to controls (C). It can be observed that there is a more significant number of species in the children recovered from anemia compared to the children with anemia and control. In the same way, greater richness and uniformity are observed in the RA group compared to the AA and C groups. Statistical tests show differences between the species observed (p_Obs_ = 0.03) and the diversity of Shannon (p_Shan_ = 0.01). In addition, the Mann–Whitney/Kruskal–Wallis test was performed to evaluate whether the samples originated from the same distribution. Results show differences between the medians for both the observed species (H_Obs_ = 6.9) and the Shannon index (H_Shan_ = 8.2).

### 3.7. Beta Diversity of the Gut Microbiota

NMDS and ANOSIM analyses, as rank-based approaches, were applied to test differences in microbial composition between the AA, RA, and C groups. The NMDS results are shown in Figure 5. Children with AA anemia (red squares) are scattered across the quadrant, suggesting no differences in composition, and confirmed by the ANOSIM result (p_ANOSIM_ = 0.819). Next, a composition similarity test (ADONIS) was performed to obtain more precise information on whether there are differences between the sample groups. Results do not show significant differences between children with anemia, children who recovered from anemia, and controls (p_ADONIS_ = 0.154). A comparison was also made between pairs of groups, in which no significant differences are found for the pairs of groups: AA–RA (p_ANOSIM_ = 0.782), C–RA (p_ANOSIM_ = 0.081), and C_AA (p_ANOSIM_ = 0.775).

NMDS is an unconstrained distance-based ordination method that was performed with Bray–Curtis dissimilarity. The dots represent samples. Samples that are more similar to each other are sorted closer together. In Figure 5, children with anemia are represented as red squares, children recovered from anemia with blue triangles, and controls as green dots. The groups do not show significant differences in the similarity tested by ANOSIM (p_ANOSIM_ = 0.819) nor by ADONIS (p_ADONIS_ = 0.154).

### 3.8. LEfSe Test for Biomarkers

The LEfSe test for biomarkers was used to find significantly unbalanced OTUs, which shows the strongest effects for group differentiation (Figure 6). Analysis at the OTU level uncovers one species, *Erysipelatoclostridium ramosum*, associated with the AA group belonging to the genus *Erysipelatoclostridium* (OTU_20, Figure 6A,B). At the genus level, *Agathobacter* and *Coprococcus* are detected as markers for the control group, and *Alloprevotella*, MBA03, and *Phascolarctobacterium* with high levels for the group of children recovered from anemia (Figure 6B). The family-level analysis shows elevated levels of MBA03 and *Acidaminococcaceae* for the group of children who recovered from anemia (Figure 6C). No significant differences are observed between the AA and control groups. At the phylum level, *Fibrobacterota* and *Synergistota* are detected as markers for the children who recovered from anemia, and no significant differences are observed between the AA and control groups (Figure 6D).

## 4. Discussion

This research presents the difference in microbial diversity in children from southern Peru. Three groups were identified for this study, children with anemia (AA), children who recovered from anemia (RA), and the control group (C). This study aims to find phyla, genera, or bacterial species that identify each study group. These findings are expected to allow us to identify possible biomarkers that can be used to make an early diagnosis of, or prevent, anemia in our children.

The intestinal microbiota fulfills vital functions in different biological processes directly impacting host health. Functions such as metabolism, nutrient absorption, neurocognitive development, and immune system efficiency are just a few examples of their critical role [12,13,14,15]. The complex intestinal microbiota–host relationship has made its study very important to understand the implications of its deficiency, especially in the health of children [16].

About 99% of the bacteria in the intestine are anaerobes; however, high densities of aerobic microbes are recorded in the cecum. The most dominant bacterial phyla in the human intestine are *Firmicutes, Bacteroidetes, Actinobacteria*, and *Proteobacteria* [17]. In addition, *Bacteroides* (year one), *Parabacteroides* (year two), and *Christensenellaceae* (year four) bacteria are found in full-term infants [18].

The results show that when analyzing the relative abundance of the intestinal microbiota, the control group (C) is the group that shows a higher proportion of the *Firmicutes* phylum, with the AA group being the one with the lowest proportion. However, the AA group has a higher abundance in the *Actinobacteria*, *Bacteroidetes*, and *Proteobacteria* phyla, the latter two comparable to the RA group. This result is similar to a study in Kenya, which reveals significant differences in taxa between anemic and non-anemic infants [19]. Non-anemic infants harbor lower abundances of *Prevotella* (2.0% vs. 4.5%, *p* = 0.014), a genus belonging to the phylum *Bacteroidetes*. Instead, they show higher quantities of *Actinomycetales* belonging to the phylum *Actinobacteria* (0.14% vs. 0.09%, *p* = 0.004), and *Streptococcus*, belonging to the phylum *Firmicutes* (6.3% vs. 3.9%, *p* = 0.023). Similarly, in another study by McClorry et al. [20], the microbiome is identified in children with anemia from the Peruvian jungle. In this study, the presence of *Firmicutes* (*Coprococcus, Dorea*, *Roseburia*), *Bacteroidetes* (*Barnesiellaceae*, *Odoribacteraceae*), and *Proteobacteria* (*Desulfovibrio*) are found, bacteria not found in children without anemia. These results show that the low bacterial presence of the *Firmicutes* phylum indicates anemia in children, which is in good agreement with other studies [21,22].

Regarding the Actinobacteria phylum, this phylum presents a higher relative abundance in the AA group, and lower for the RA group. Contrasting with our study, in research carried out by Muleviciene et al. [23], the microbiota of ten children with IDA and ten control children are identified. Their results show that children with IDA have increased *Bacteroidetes* and *Proteobacteria*, and a lower amount of *Actinobacteria* and *Verrucomicrobia*, unlike control children. These results agree with those found by Jaeggi et al. [19].

The phylum *Proteobacteria* is less abundant in C than in AA and RA. However, as in the study carried out in India in children, most of them anemic, a greater abundance of this bacteria is found [24]. The presence of this phylum is associated with intestinal microbial imbalance (dysbiosis) and harmful health effects. Therefore, several studies propose it could be used as a biomarker for this disorder, or other diseases associated with the gut microbiome [25,26,27]. Furthermore, the similarity in the number of bacteria of the *Bacteroidetes* and *Proteobacteria* phyla in AA and RA children indicates that the children present an imbalance in the microbiota due to anemic conditions.

On the other hand, regarding the phyla with a lower abundance, our results show that, in the AA group, there is a greater proportion of the *Chloroflexi* phylum. In contrast, in the RA group, it is the *Cyanobacteria* phylum. Furthermore, *chloroflexi* bacterium is present in the intestinal microbiota of non-Westernized children in a study conducted in Mexico through sequencing the 16S rRNA gene [28]. Moreover, the *Verrucomicrobiota* phylum is present in AA and RA groups. A study in Canadian children with selenium in the blood shows a high relative abundance of *Verrucomicrobiota*. This bacterium is ubiquitous in the human intestine and can live in association with eukaryotes [29]. Finally, in the RA group, *Cyanobacteria* appear. A study conducted in Champaign-Urbana, US, in children aged 4 to 8 years with intake characterized by cereals, dairy, legumes, nuts, and seeds, unlike another group that receive an intake of fish, refined carbohydrates, protein foods, fruits, vegetables, juices, and sugary drinks, shows a high abundance of *Cyanobacteria* [30].

From the analysis of the number of shared OTUs (Appendix A), it is observed that the RA group has the highest percentage of shared OTUs (17.6%), and the lowest is the AA group (5.4%). When analyzing the relative abundance of these two groups, the results show that the *Clostridia* class (*p* = 0.027) and the orders *Peptostreptococcales-Tissierellales* (*p* = 0.005) and *Clostridiales* (*p* = 0.017), belonging to the *Firmicutes* phylum, have significant differences (Figure 3). The *Clostridia* spp. and the *Peptostreptococcales-Tissierellales* order are 2.5 and 3.1 times higher for the RA group, respectively. The increase in these microbiotas indicates recovery pictures in children. In this sense, a study by Lopetuso et al. points out the importance of some species of the *Clostridia* class in human health, especially in the immune system [31]. In particular, the gut commensal *Clostridia* bacteria can represent up to 40% of the total bacteria of the intestinal microbiota. In contrast, the AA group shows a greater abundance of the Clostridiales genus (3.2 times higher), which could be interpreted as evidence of anemic symptoms. However, more robust analyzes are necessary to identify the bacterial composition, as there are still several bacterial species to be discovered, as shown by the rarefaction curves (Appendix A).

Our results show that the species *Erysipelatoclostridium ramosum* has a significant taxonomic difference for the AA group. Furthermore, studies show that this bacterium is associated with human metabolic syndrome and mice-induced obesity [32,33,34]. As for the *Erysipelatoclostridium* genus, studies show that its abundance is significantly higher in down-regulated immune systems [35,36,37]. Regarding the RA group, the genera that show the greatest taxonomic difference are *Alloprevotella*, *MBA03*, and *Phascolarctobacterium*. In the case of the *Alloprevotella* genus, its high abundance in the intestine is related to the healthy growth and physical health of the host [38,39], as well as to the recovery of metabolism in children with epilepsy [40]. The presence of bacteria of the *Phascolarctobacterium* genus prevents the growth of *Clostridioides difficile* bacteria [41], a highly infective agent in children [42]. In addition, a decrease in its abundance is found in children with chronic pancreatitis. In the case of the *MBA03* genus, there is not much information regarding its presence in the intestinal microbiota [43].

Although our analyzes were performed with a small sample size, we believe that the findings presented in this work will open new perspectives in research on the importance of the gut microbiota in the health of our children. However, further robust and in-depth studies must be conducted to evaluate microbial diversity and understand its impact on anemia and various childhood diseases.

## Figures and Tables

**Figure 1 children-09-01615-f001:**
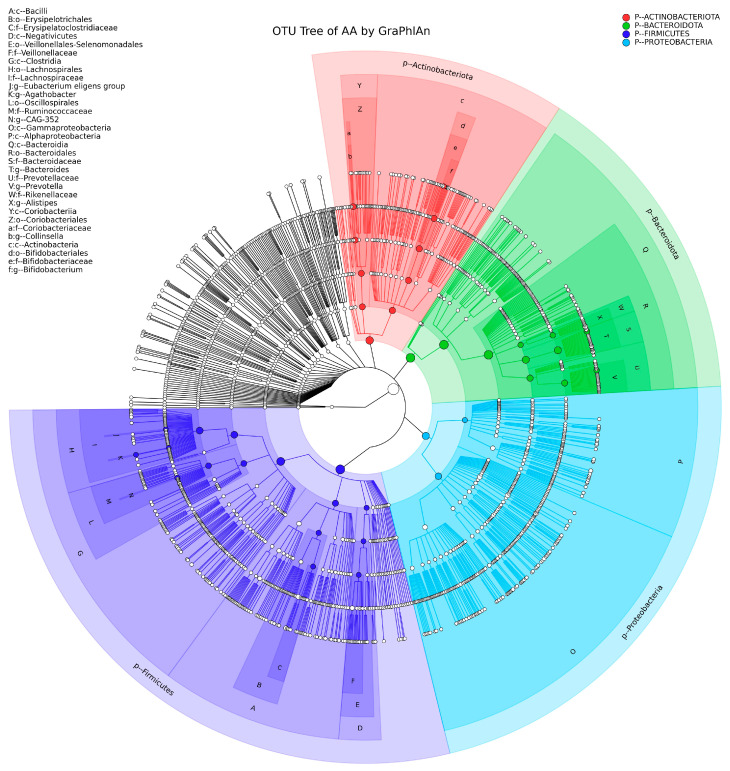
GraPhIAn of the group of children with anemia. The red color represents *Actinobacteriota*, green *Bacteroidota*, light blue *Proteobacteria*, and purple *Firmicutes*. *Firmicutes* are the most representative phylum of the sample of children with anemia. The solid circles represent the 40 most abundant species; on the far left of the figure is the species nomenclature.

**Figure 2 children-09-01615-f002:**
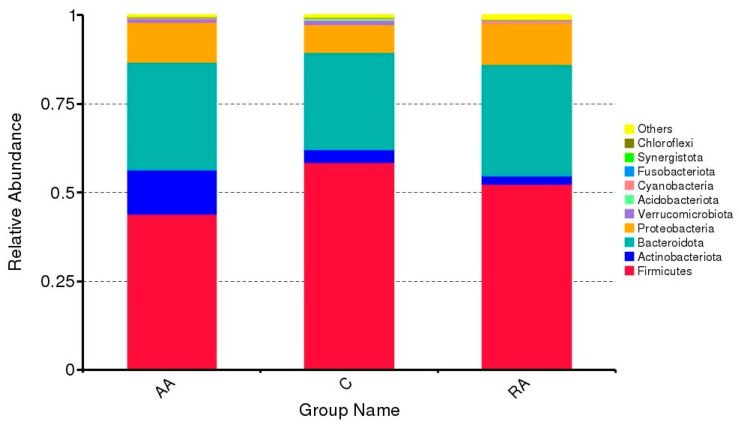
Relative abundance in the groups of children with anemia (AA), children recovered from anemia (RA), and control children (C). Sample AA shows a higher abundance of *Actinobacteriota*, while C shows a higher abundance of *Firmicutes* and RA of *Bacteroidota*. About 90% of the microbial composition is in the three groups is *Firmicutes*, *Actinobacteriota*, and *Bacteroidota*.

**Figure 3 children-09-01615-f003:**
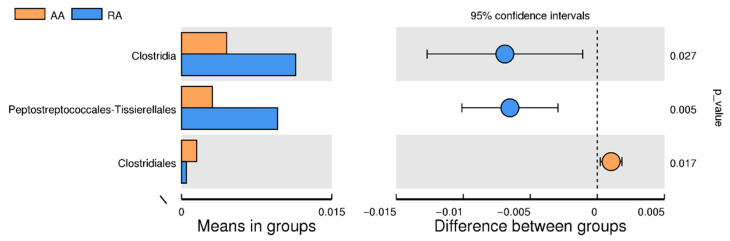
Between-group *t*-test analysis. The orange color shows the group AA, and the blue color the group R.

**Figure 4 children-09-01615-f004:**
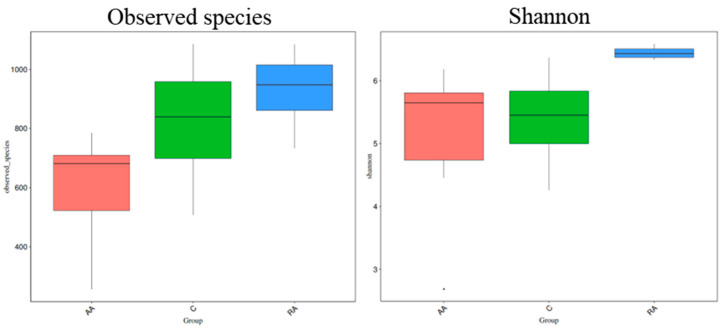
Alpha diversity, as measured by observed species and the Shannon diversity index, is plotted for children with anemia AA (red), children recovered from anemia RA (green), and controls C (blue). The line inside the box represents the median, and the rhombus the mean. The whiskers (vertical lines) represent the lowest and highest values within the interquartile range. Outliers, as well as individual sample values, are shown as dots. Statistical tests show differences for the species observed (p_Obs_ = 0.03; H_Obs_ = 6.9), as well as for Shannon diversity (p_Shan_ = 0.01; H_Shan_ = 8.2).

**Figure 5 children-09-01615-f005:**
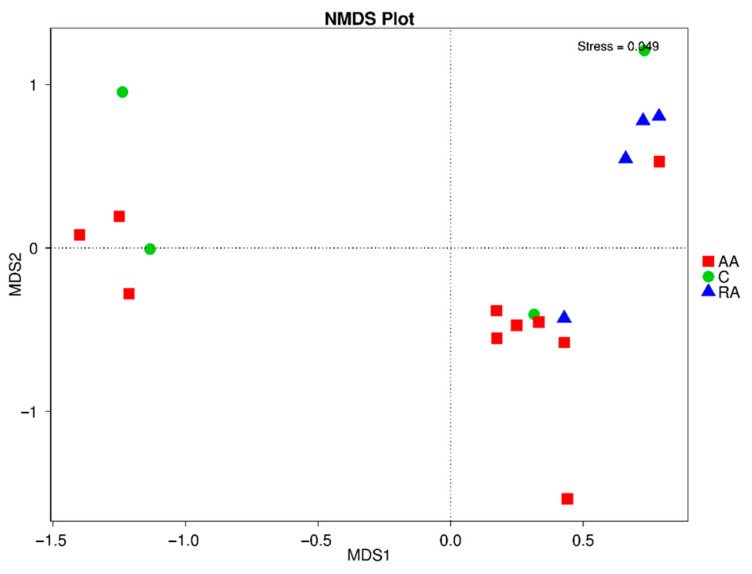
Graph of the non-metric multidimensional scaling (NMDS) analysis performed on the samples of AA, RA, and C groups.

**Figure 6 children-09-01615-f006:**
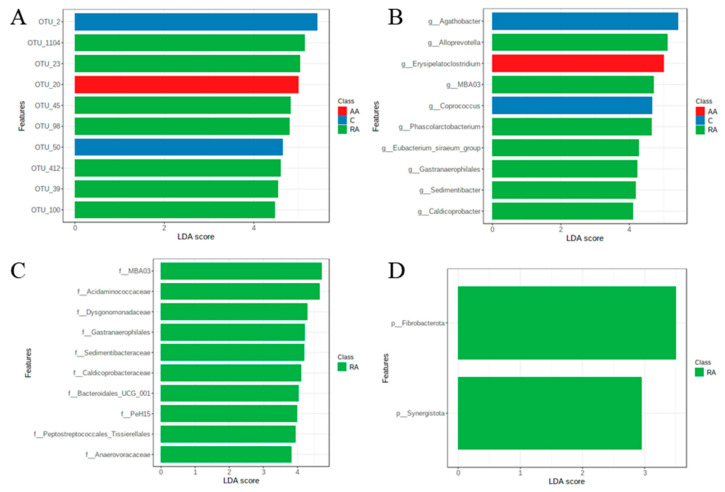
Results of LDA effect size analysis (LEfSe) of children with anemia (AA) and children recovered from anemia (RA) compared with healthy controls (C). The LEfSe analysis finds taxa that are significantly more abundant in a group, the size of the bar representing the effect size of the taxa in the particular group. Children with anemia (AA) are indicated in red, children recovered from anemia (RA) in green, and healthy controls (C) in blue. (**A**) OTU level (97% similarity), (**B**) genus level, (**C**) family level, and (**D**) phylum level. The threshold *p*-value is 0.05. The log LDA score threshold for discriminant characteristics is set at 2.0.

## Data Availability

Not applicable.

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
