# Peer review of "Intestinal Microbiota in Children with Anemia in Southern Peru through Next-Generation Sequencing Technology"

_children, 2022, doi:10.3390/children9111615_

Round 1

Reviewer 1 Report

The investigation addresses a relevant topic about interaction between intestinal microbiota and pathological conditions, which is well known.

The sample size of the population is very small and divided in 3 groups leaving low consistency to the results obtained.

There was no paralel investigation on other conditions or treatments which brings additional heterogeneity to the interpretation of results, as observed in the cited ref 11.

The observation of associations does not provide causality or even direct relationship.

The last sentences (A high quantity of this phylum suggests showing a behavior as a marker of dysbiosis (intestinal microbial imbalance) and has been associated with harmful health outcomes. Additionally, in the AA group, the Chloroflexi phylum is present, unlike the other groups, while the RA group presents the Cyanobacteria 218 phylum), which also needs some grammatical correction, is merely observational e broad, not providing any consistent conclusion from the work done or proposing any relevant way forward.

Overall I think the work, of some merit, needs some proper context and relevant interpretation that may be useful for the reader.

Reviewer 2 Report

Diaz-Rodriguez et al., characterized the gut microbiome of children with and without anemia. This manuscript is well polished and has the potential to impact the biomedical field. Specific comments are listed below.

1. The manuscript can start with the sentence "Recent studies indicate that iron deficiency (ID) [4] and iron deficiency anemia (IDA) cause unfavorable changes in the intestinal microbiota [5]." Introduction before this sentence is not necessary. However, more elaboration of the introduction regarding anemia and gut microbiome would strengthen the manuscript.

2. The statistics description is completely missing as well as the information on whether sequences are publicly available or not.

3. Figure 3 and Figure5 are not necessary to be in the main figure. These can be supplementary figures or replaced with descriptions, especially for figure 3.

4. Figures 4 and 7 should be displayed as a single figure in panels A and B. This is good because the authors used 2 different approaches. However, Figures 4 and 7 should be revisited using the same taxonomic level. Comparing Order to Family, or comparing Family to Species are as if comparing apples and oranges.

5. Lastly, the discussion section needs a more in-depth discussion about how the authors' finding from this study connects to the function of anemia and the future direction.

Round 2

Reviewer 1 Report

I am happy with reply from the authors and modifications in the manuscript. However, please note and correct that the legend of figure 5 is written in Spanish.

Reviewer 2 Report

1. My original comment #2 (#2: The statistics description is completely missing as well as the information on whether sequences are publicly available or not) is still not addressed. Next to the 2.3 bioinformatics analysis, the description of which statistical tests were applied is needed.

2. Results subtitles should be the findings, not the software names or figure format names used.

3. Correct fig43 to fig 3

4. My original comment #5 (#5: Lastly, the discussion section needs a more in-depth discussion about how the authors' finding
from this study connects to the function of anemia and the future direction.) has not been addressed by the authors sufficiently.

Round 3

Reviewer 2 Report

The manuscript has improved and appropriately addressed all my questions and comments.